# Molecular Characterization of Odorant-Binding Protein Genes Associated with Host-Seeking Behavior in *Oides leucomelaena*

**DOI:** 10.3390/ijms25179436

**Published:** 2024-08-30

**Authors:** Ning Zhao, Kai Li, Huifen Ma, Lianrong Hu, Yingxue Yang, Ling Liu

**Affiliations:** 1College of Biological Science and Food Engineering, Southwest Forestry University, Kunming 650224, China; lijiangzhn@swfu.edu.cn (N.Z.); likaiswfu@163.com (K.L.); yangyingxue1@sona.com (Y.Y.); 2Yunnan Academy of Forestry and Grassland, Kunming 650224, China; mahuifen@yafg.ac.cn (H.M.); hulianrong@yafg.ac.cn (L.H.)

**Keywords:** *Oides leucomelaena*, odorant-binding protein, plant volatile organic compounds, molecular docking

## Abstract

The identification of odorant-binding proteins (OBPs) involved in host location by *Oides leucomelaena* (*O. leucomelaena* Weise, 1922, Coleoptera, Galerucinae) is significant for its biological control. Tools in the NCBI database were used to compare and analyze the transcriptome sequences of *O. leucomelaena* with OBP and other chemosensory-related proteins of other Coleoptera insects. Subsequently, MEGA7 was utilized for OBP sequence alignment and the construction of a phylogenetic tree, combined with expression profiling to screen for candidate antennae-specific OBPs. In addition, fumigation experiments with star anise volatiles were conducted to assess the antennae specificity of the candidate OBPs. Finally, molecular docking was employed to speculate on the binding potential of antennae-specific OBPs with star anise volatiles. The study identified 42 candidate OBPs, 8 chemosensory proteins and 27 receptors. OleuOBP3, OleuOBP5, and OleuOBP6 were identified as classic OBP family members specific to the antennae, which was confirmed by volatile fumigation experiments. Molecular docking ultimately clarified that OleuOBP3, OleuOBP5, and OleuOBP6 all exhibit a high affinity for β-caryophyllene among the star anise volatiles. We successfully obtained three antennae-specific OBPs from *O. leucomelaena* and determined their high-affinity volatiles, providing a theoretical basis for the development of attractants in subsequent stages.

## 1. Introduction

Star anise (*Illicium verum* Hook. F.) is a species belonging to the Anise genus of the Magnoliaceae family. It is renowned for its diverse medicinal attributes such as antioxidant, antibacterial, anti-inflammatory, insecticidal, and antiviral effects [1]. This plant plays a significant role in East Asian traditional medicine, with China recognized as its primary region of cultivation [2,3]. However, the growth of star anise is highly susceptible to *Oides leucomelaena* (*O. leucomelaena* Weise, 1922, Coleoptera, Galerucinae), which causes huge losses to the regional economy [3]. It is reported that the larvae and adults of *O. leucomelaena* feed on the leaves and young shoots of anise trees, causing withering of branches, lack of fruits, and even death of the whole plant [4,5]. At present, the control of this insect mainly relies on chemical control, which is fast but harmful to the environment and negatively impacts natural enemies, which is not conducive to the sustainable control of the insect pest [6]. *O. leucomelaena* inhabits a complex and chemically diverse environment, and essential to its survival is the need to recognize host plant odors in the environment in search of hosts. Therefore, interfering with the search for hosts by *O. leucomelaena* may be a viable option.

During coevolution with their hosts, insects have developed a unique olfactory system enabling them to distinguish volatile substances in host plants from other environmental odors. Insect olfactory systems are known for their remarkable sensitivity and the ability to integrate odorant blends through distributed specificity of receptor tuning profiles. The olfactory systems of insect perception of host plants may help reveal the co-evolutionary relationship between insects and plants, and provide a theoretical basis for the development of ecological pest control techniques [7,8]. And insects’ perception of host plant volatiles usually depends on their chemoreceptors. Odorant-binding proteins (OBPs) are essential molecules for host localization. Previous studies have shown that the process by which insects perceive external odors mainly involves the binding, release and inactivation of odor molecules [9]. First, hydrophobic odor molecules enter the hemolymph through the micropores of the sensors as well as the pore microtubules and are then bound by OBPs and chemosensory proteins (CSPs). The stable and compact structures of OBPs and CSPs make them multifunctional soluble proteins relevant for signal transduction of small-molecule hydrophobic compounds such as pheromones and odors, as the hydrophobic odor molecules become soluble due to the binding of the hydrophobic odor molecules to the OBPs. Subsequently, OBPs transport odor molecules to the dendritic membranes of olfactory receptor neurons to activate receptors. Afterwards, the receptors convert the chemical signals into electrophysiological signals, which are transmitted via axonal nerves to the central nervous system, where the brain generates commands to the insect based on the signals, guiding its behavior [10,11]. The research value of OBPs as the first step in insect odor recognition cannot be overstated.

Insect OBPs are small, globular, water-soluble proteins whose protein sequences include highly conserved cysteines, which in turn have a specific number of amino acid residues between them. For example, the classical OBP consists of six conserved cysteine residues, of which Coleoptera are divided into the following two patterns: C1-X23-44-C2-X3-C3-X36-43-C4-X8-12-C5-X8-C6 and C1-X21-68-C2-X3-C3-X21-46-C4-X8-28-C5-X8-9-C6 [12,13]. The classical OBP’s three-dimensional structure consists of six α-helical structural domains that form a hydrophobic cavity. In addition, the six conserved cysteines form three interlocking disulfide bonds and fold to form a tight and stable hydrophobic binding cavity, which increases the structural stability of the OBP to a certain extent [14]. The stable sequence structure of OBPs plays an important role in maintaining their functions. Studies on odor-binding proteins of Coleoptera have mainly focused on bark beetles, and few studies on leaf beetles have been reported [15,16]. As a pest that is very harmful to *O. leucomelaena*, green science must be urgently used to control *O. leucomelaena*. In the last decade, emerging evidence suggests that OBPs play an important role in decontaminating the surrounding perceptron space from harmful xenobiotics, including plant volatiles and pesticides, which may facilitate adaptations to xenobiotics such as host localization, acclimation and pesticide resistance [8]. In this study, we identified the chemosensory-related proteins of *O. leucomelaena* by sequencing the transcriptome of different tissue parts of *O. leucomelaena*, and focused on the correlation analysis of OBPs, with a view to clarifying the key role of the antennal-specific OBPs of *O. leucomelaena* in the host localization on the octocarp, and to provide the theoretical basis for the subsequent biocontrol of *O. leucomelaena*.

## 2. Results

### 2.1. Library Assembly

As a result of sequencing Illumina MiSeq-paired libraries of male and female tissues of *O. leucomelaena*, we obtained a total of 749,085,848 clean reads, a GC percentage of 34.76–42.88%, and Q20 percentage of 96.86–98.08% (Appendix A). The reads from the paired libraries were assembled in 374,907 sequences, with a mean length of 855 bp and N50 length of 1134 bp. After cleaning redundancies, 171,155 unigenes were obtained, with a mean length of 802 bp and N50 of 1004 bp (Appendix A). All read data are available in the NCBI BioProject database under the project ID PRJNA1123008.

### 2.2. Transcript Annotation

The Gene Ontology (GO) and Kyoto Encyclopedia of Genes and Genome (KEGG) databases were used for transcript annotations, with 53,017 unigenes in at least one of the three GO terms (Appendix A). Biological processes were the most representative corresponding to cellular processes and metabolic processes. A total of 12 terms in molecular functions were found, with binding being the most representative subcategory, followed by catalytic activity, transport, and biological regulation. Then, five categories were assigned cellular components, with the most represented subcategories being cellular anatomical entity and intracellular (Figure 1A). Meanwhile, the annotations of unigenes in KEGG showed that the most enriched entries for genetic information processing contain 8421 unigenes, and 4002 unigenes were enriched in the signaling and cellular processes term (Appendix A). The unigenes were enriched in the sensory system and nervous system entries (Figure 1B). Currently, there is limited research on the chemosensory-related proteins of *O. leucomelaena*, but through unigene enrichment analysis, it is not difficult to find that there are chemosensory-related genes within its body that are in urgent need of exploration.

### 2.3. Chemosensory-Related Proteins

We identified 42 unigenes encoding OBPs with an open reading frame (ORF) in length from 49 to 200 amino acids (aa) in the *O. leucomelaena*. The BLAST matches of these OBPs with other OBPs of insects with 27.84–80.60% percentage identity, mainly with a member of the same genera *Galeruca daurica* (Joannis, 1865), followed by others Coleoptera Curculionidae: *Diabrotica virgifera virgifera* (LeConte,1868), *Apriona germarii* (Hope, 1831), and *Pyrrhalta aenescens* (Fairmaire, 1878), among which OleuOBP3, OleuOBP5, and OleuOBP6 showed greater than 60% homology similarity (Appendix A).

In addition to OBP, a total of eight CSPs were identified, with sequence similarity between 51.14 and 78.79% and ORF lengths between 83 and 250 aa, and most of the CSPs have high homology with *Ophraella communa* (LeSage, 1986). In addition, we also identified 19 odorant receptors (ORs), three ionotropic receptors (IRs), and five gustatory receptors (GRs). The similarity of ORs was between 26.97 and 90.81%, and both GRs and IRs have high similarity with the related receptors of *Diorhabda carinulata* (Desbrochers, 1869) (Appendix A).

Expression analysis of chemosensory-related proteins obtained from the *O. leucomelaena* heatmap showed that OleuOBP1, OleuOBP2, OleuOBP3, OleuOBP5, OleuOBP6, OleuOBP9, OleuOBP17 and OleuOBP29 had clear antennal specificity, and OBP26 was highly expressed in all tissues (Figure 2A). OleuCSP1, as well as most of OleuOR, were highly expressed in the antenna (Figure 2B,C). In addition, the IR of *O. leucomelaena* was expressed in various tissues, while OleuGR exhibits a broad pattern of low expression (Appendix A). Insect antennae, as an important sensory organ where chemoreceptors are located, have a greater research value, and the antennal-specific OleuOBPs in our study were used as the focus of the study to carry out the next step of analysis.

### 2.4. Sequence Comparison and Phylogenetic Tree Analysis of OBPs

OBP sequence comparisons of *O. leucomelaena* revealed that two OBP subfamilies were annotated in *O. leucomelaena* based on the conserved cysteine (Cys) pattern. Seven correspond to the classic OBPs (OleuOBP3, OleuOBP5, OleuOBP6, OleuOBP7, OleuOBP13, OleuOBP19 and OleuOBP37, with six conserved Cys (Figure 3), and twenty-seven assigned to the Minus-C OBP, with four conserved Cys and the absence of C2 and C5 (Appendix A).

The OBP sequences of *O. leucomelaena* and other Coleoptera insects were integrated for the construction of a phylogenetic tree of homologous species and the comparative analysis removed the shorter four OleuOBPs (OleuOBP8, OleuOBP9, OleuOBP17, OleuOBP39) sequences and retained 38 sequences OleuOBPs with 82 sequences of other insect OBP for the construction of phylogenetic tree (Figure 4). The phylogenetic tree showed that the OBPs of *O. leucomelaena* were classified into two major families; this result was consistent with the result of the comparison analysis. OleuOBP3 and OleuOBP5 as classical OBP families have some homology with those of *Diorhabda carinulata* (bootstrap > 0.73). In addition, OleuOBP6 was homologous to the *Tribolium castaneum* (Herbst, 1797) OBP sequence (bootstrap > 0.59).

### 2.5. Antennal-Specific OBP

Olfactory fumigation experiments indicated that under the treatment condition of 70 uL of the mixture, the mortality rates of the three groups of insects were 53%, 46%, and 56%, respectively, reaching the LC50 for *O. leucomelaena* (Figure 5). The qRT-PCR results of the antennae showed that the expression levels of OleuOBP3, OleuOBP5, and OleuOBP6 in the treatment group were significantly higher than those in the control group. In contrast, qRT-PCR results from other tissues showed no expression of OleuOBP3, OleuOBP5, and OleuOBP6 in either group, further confirming OleuOBP3, OleuOBP5, and OleuOBP6 as antennae-specific OBPs in *O. leucomelaena* (Figure 6A–C).

### 2.6. β-Caryophyllene Is a Molecule with High Affinity for Antennae-Specific Proteins

Molecular docking simulations were conducted using the four compounds from the fumigation experiment with antennae-specific OBPs. The results of the molecular docking indicated that OleuOBP3, OleuOBP5, and OleuOBP6 all exhibited high affinity for the four compounds, with β-caryophyllene showing the strongest binding affinity for all three OleuOBPs (binding energy < −21 kJ/mol, Table 1). Visualizing the protein–ligand docking structures revealed the following binding active sites: β-caryophyllene bound to OleuOBP3 at site TYR-105 (Figure 7A), to OleuOBP5 at site TYR-79 and HIS-33 (Figure 7B), and to OleuOBP6 at site TRP-116 (Figure 7C).

## 3. Discussion

In this study, transcriptome sequencing of various tissues of *O. leucomelaena* was performed to identify OBPs associated with beetle host location. Other than that, it was only clear that β-caryophyllene from anise volatiles is a molecule with high affinity for antennal-specific OBPs of *O. leucomelaena*.

After analyzing the transcriptomes of different tissues of *O. leucomelaena*, it was found that the annotated unigenes were enriched in pathways related to genetics and signal processing. Kang Le and others identified insect-specific proteins through the analysis of genomic information from insects with different metamorphosis types [16]. These proteins are related to environmental adaptation and information exchange, demonstrating the uniqueness of insects in the process of adapting to the environment and evolving [17,18]. The study found that most insect-specific proteins have a low mutation rate during the evolutionary process, indicating that insect genetic information plays an important role in providing a stable protein composition [16,19]. At the same time, the genetic diversity of insects is considered to be the result of their adaptation to the environment and long-term evolution, and insect-specific proteins are an important characteristic that distinguishes insect species differentiation and behavioral habits from other organisms [20,21]. In addition, insects control signal transduction to perceive changes in the external environment through specific proteins, such as plant volatiles, temperature changes, humidity, etc. The genes of these proteins are crucial for insects to find food sources, choose breeding sites, and avoid adverse environments [22].

The number of chemosensory genes varies among insect taxa due to differences in evolutionary processes, including the acquisition and loss of genes and genetic mutations. Such differences among insects are thought to be related to specific lifestyles and adaptations to the environment and may ultimately lead to species divergence. The amount of OBP genes in an insect genome can vary greatly among species, ranging from as low as 7 in *Ceratosolen solmsi* to as high as 111 in *Aedes aegypti* [8]. Predictions based on literature reports combined with our results suggest that host-specific insects will possess more chemosensory-related genes compared to polyphagous species [7,23,24]. In this study, a total of 42 OBPs, 19 CSPs, and 27 receptors were identified. Currently, there is limited research on OBPs in Chrysomelidae insects, which provides few references. The research by Yinliang Wang and colleagues identified 15 OBPs in *Ambrostoma quadriimpressum* [22]. L Li obtained 29 candidate OBPs in the study of *Galeruca daurica* [25]. Meanwhile, the homology comparisons in our results were not entirely satisfactory, which partly explains the differences in the number of gene identification results. Additionally, the specific living environments of various insects in the Chrysomelidae family are also one of the reasons for the differences in the number of chemosensory genes. In contrast, reports of chemosensory-associated proteins against other insects are not uncommon. For example, Qing-Feng Tang and his team identified 41 candidate OBPs in their study of *Sitophilus zeamais*, which were divided into the dimeric OBP subfamily, minus-C OBP subfamily, and classic OBP subfamily, with most OBPs being highly expressed in the antennae and head. Martin N Andersson and his colleagues obtained 86 ORs, 60 GRs, 57 IRs, 36 OBPs, and 11 CSPs in their research on *Dendroctonus ponderosae*, and 47 ORs, 30 GRs, 31 IRs, 12 OBPs, and 14 CSPs in their research on *Agrilus planipennis* [26]. As a polyphagous insect, *Agrilus planipennis* has a reduced gene pool compared to Chrysomelidae insects like *Dendroctonus ponderosae* and *O. leucomelaena*. It is hypothesized that there is a correlation between the content of chemosensory genes in beetles and their host specificity [27].

In the analysis of the expression profile of *O. leucomelaena*, it was found that the expression of chemosensory-related genes has certain tissue specificity, with OBPs showing clear tissue specificity (antennal specificity). Of course, some chemosensory-related genes (OBPs, CSPs, ORs, GRs, IRs) are expressed in a wide range of tissues. In recent years, the identification and study of insect chemosensory-related genes have shown that these genes have a broad tissue expression pattern [28,29]. However, most OBPs in insects exhibit antennal specificity [30,31]. The antennae contain a large number of chemoreceptors, and the discovery and functional analysis of antennal-specific OBPs indicate that they are key proteins for binding odor substances and transmitting information in the chemosensory system [28,32]. The value of clarifying the antennal-specific OBP is that it can be used as a basis to explore whether it plays a role in insect host localization. The sequence alignment of OBPs revealed that the existing OBP of the *O. leucomelaena* is mainly divided into the classic OBP family and the minus-C OBP family. In the construction of the phylogenetic tree for the classic OBP family, it is clear that species with closer phylogenetic relationships have higher bootstrap values and higher sequence homology. This suggests that there is a certain degree of variation in the evolutionary process of insects, while also indicating that sequences among closely related species retain conservation [33,34].

In our study, it was confirmed that star anise volatiles significantly affect the OBPs of *O. leucomelaena*. Under conditions treated with volatiles, the expression of antennal OleuOBP3, OleuOBP5, and OleuOBP6 in *O. leucomelaena* was significantly increased, further establishing the key value of these three proteins in the host location for *O. leucomelaena*. Molecular docking experiments performed on these three OBPs also confirmed their high affinity for important volatile components in star anise. Among them, β-caryophyllene has the highest affinity for OBPs. β-caryophyllene is the main chemical component in star anise and is synthesized in the plant as part of the plant’s defense mechanism to help defend against insect pests [35]. Research indicates that β-caryophyllene can influence the behavior of pests, for instance, by disrupting their feeding, reproduction, or the ability to locate host plants. Given its potential impact on pests, β-caryophyllene is being investigated as an alternative or supplementary approach in pest management, especially in the search for environmentally friendly pest control strategies [36,37]. β-caryophyllene may exhibit synergistic effects with other compounds, such as other plant volatiles or pheromones, potentially enhancing its efficacy in pest control and its efficacy may be a new idea for *O. leucomelaena* prevention and treatment [35].

## 4. Materials and Methods

### 4.1. Insect Collection

In June 2023, we collected adult *O. leucomelaena* in Funing County, Wenshan Prefecture, Yunnan Province, and separated and dissected the male and female adult insects in the Biological Science laboratory. We isolated and obtained the antennae, head, thorax, abdomen, legs, and wings of *O. leucomelaena* and preserved them in liquid nitrogen for subsequent RNA sequencing. In May 2024, we collected adult *O. leucomelaena* in Funing County, Wenshan Prefecture, Yunnan Province, China, for use in olfactory fumigation experiments with *O. leucomelaena* involving anise-related compounds.

### 4.2. RNA Sequencing and Annotation

We utilized the NEBNext UltraTM RNA Library Prep Kit (New England Biolabs, Ipswich, MA, USA) to create sequencing libraries, employing AMPure XP beads (Beckman Coulter, Beverly, MA, USA) for the purification of cDNA fragments and PCR to enrich them. Subsequently, these cDNA libraries were prepared for paired-end sequencing on the Illumina Novaseq6000 platform at Novogene Bioinformatics Technology Co., Ltd. (Beijing, China). Initial raw sequence reads underwent meticulous quality control to remove substandard reads based on base calling accuracy. The clean reads were then assembled de novo using Trinity (version 2.4.0) and clustered to streamline the transcriptome data, eliminating redundant data with the aid of Corset (version 1.0.5) [38]. For functional annotation, we leveraged the GO and KEGG databases. GO term enrichment was executed with the blast2go software suite (version 2.5), while pathway analysis within the KEGG database was conducted using hmmscan from the HMMER 3 package [39,40]. Read counts for each mapped gene were normalized by TMM and used to calculate gene expression levels according to the FPKM (fragments per kilobase of transcript sequence per millions base) method.

### 4.3. Gene Identification and Sequence Analysis

To identify candidate chemosensory-related genes from *O. leucomelaena*, chemosensory-related gene families from other coleoptera species were selected as queries to search the new stand-alone transcriptome of this beetle. TBLASTN was used to search and identify candidate chemosensory-related genes against the *O. leucomelaena* transcriptome, with an E-value cutoff of 1 × 10^−5^ [41]. Further, these identified genes were verified using TBLASTX against the NCBI non-redundant protein sequence database. ORFs were identified using the ORF Finder in NCBI (https://www.ncbi.nlm.nih.gov/orffinder/ (accessed on 10 January 2024)). In the set of trees, a multiple sequence alignment was performed using the Muscle method in MEGA7.0 [42]. An Neighbor-Joining tree of OBP was constructed by iTOL v5 [43]. Accession numbers of all protein sequences from other coleoptera species used in the phylogenetic analysis are listed in Appendix A.

### 4.4. Compound Fumigation Experiment

Laboratory-cultured unmedicated anise leaves were fed to *O. leucomelaena* for 3 days and then grouped for chemical fumigation experiments. Using fumigation toxicity tests, 30, 50, 70, 90, and 110 μL of the mixture (solvent: n-hexane, (Z)-anethole 900 μg/mL, β-caryophyllene 30 μg/mL, 4-allylanisole 3 μg/mL, and 3-carene 2 μg/mL, in a ratio of 653:23:2:1) were added to culture dishes (150 × 25 mm), respectively. Each group consisted of 30 adult *O. leucomelaena* (15 females, 15 males) placed in culture dishes lined with filter paper. The treated culture dishes were placed in a refrigerator at 6 °C for one hour. After 1 h, the dishes were taken out of the refrigerator and left to stand for 30 min. The *O. leucomelaena* was removed for observation and examined under an in stereo microscope (Nikon AZ100, Tokyo, Japan) with a clean brush and tweezers; no response to touch was considered dead. The survival rate of *O. leucomelaena* was then reported, and the mortality rate was calculated.

In total, 30 adult *O. leucomelaena* were placed in culture dishes (150 × 25 mm) lined with filter paper. Dry filter paper was placed on top of the culture dish. The compound (lethal concentration 50, LC50) was added to the top filter paper, and the culture dish was immediately sealed with plastic wrap. Each group was replicated three times, and the control group was added with an equal volume of n-hexane. The treated culture dishes were placed in a refrigerator at 6 °C for one hour. After treatment with the compound, the survival of *O. leucomelaena* was observed under a binocular microscope. Live beetles were selected for dissection, antennae, head, thorax, abdomen, foot, and wings were taken for RNA extraction, and the targeted OBP genes were analyzed by qRT-PCR.

### 4.5. Specific Expression of OBP Genes

Total RNA was isolated from samples using the TRIzol reagent (Invitrogen, Carlsbad, CA, USA). Subsequently, the synthesis of the first-strand cDNA was carried out with the aid of the PrimeScript™ RT reagent Kit (TaKaRa, Tokyo, Japan), adhering to the manufacturer’s protocol. To evaluate the tissue-specific expression profiles of OleuOBPs, quantitative real-time PCR (qRT-PCR) was utilized. This process was conducted on the ABI PRISM 7500 platform (Applied Biosystems, Foster City, CA, USA). The PCR mix included 10 μL of SYBR Premix Ex TaqTM II (TaKaRa, Tokyo, Japan), 1 μL each of the primers at a concentration of 10 μmol/L, 2 μL of the cDNA template, 0.4 μL of the Rox Reference Dye II, and 5.6 μL of nuclease-free water to complete the volume. The thermal cycling conditions initiated with a denaturation step at 95 °C for 30 s, followed by 40 cycles of denaturation at 95 °C for 5 s and annealing/extension at 60 °C for 34 s.

The primers for OleuOBPs were designed using GenScript online (Appendix A). The β-actin gene of *O. leucomelaena* was used as the reference gene. All samples were tested with three biological replicates. Relative quantification was performed using the 2^−ΔΔCT^ method. The differences in the transcription levels of the two OBPs groups were compared using Tukey’s test, and the bar chart shows the differences in the mRNA expression levels of OleuOBPs among the different groups.

### 4.6. Molecular Docking

The three-dimensional structure of OBP, more than 30% homology with the OBP templates in protein database,1 was modeled by program SWISS-MODEL [44] (Appendix A). AutoDock 4 (version 1.5.7) was selected to analyze the binding mode between the OBPs and compounds of *O. leucomelaena* with the default parameters [45]. The top ranked binding mode was evaluated according to the docking score, and visually analyzed by PyMOL (version 2.5.4; http://www.pymol.org/, accessed on 28 May 2024).

## 5. Conclusions

Our research has identified candidate chemosensory-related genes in *O. leucomelaena* and clarified three antennae-specific proteins, providing a theoretical basis for the biological control of this pest. However, the study of the chemosensory system in the *O. leucomelaena* is not comprehensive. Future research will focus on exploring the molecular mechanisms of the chemosensory system in the *O. leucomelaena*, with the aim of developing new control methods for this species.

## Figures and Tables

**Figure 1 ijms-25-09436-f001:**
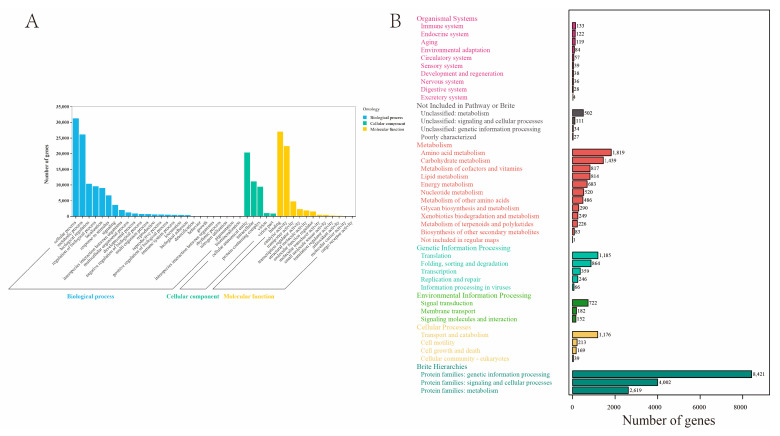
Gene Ontology (**A**) and Kyoto Encyclopedia of Genes and Genomes (**B**) results.

**Figure 2 ijms-25-09436-f002:**
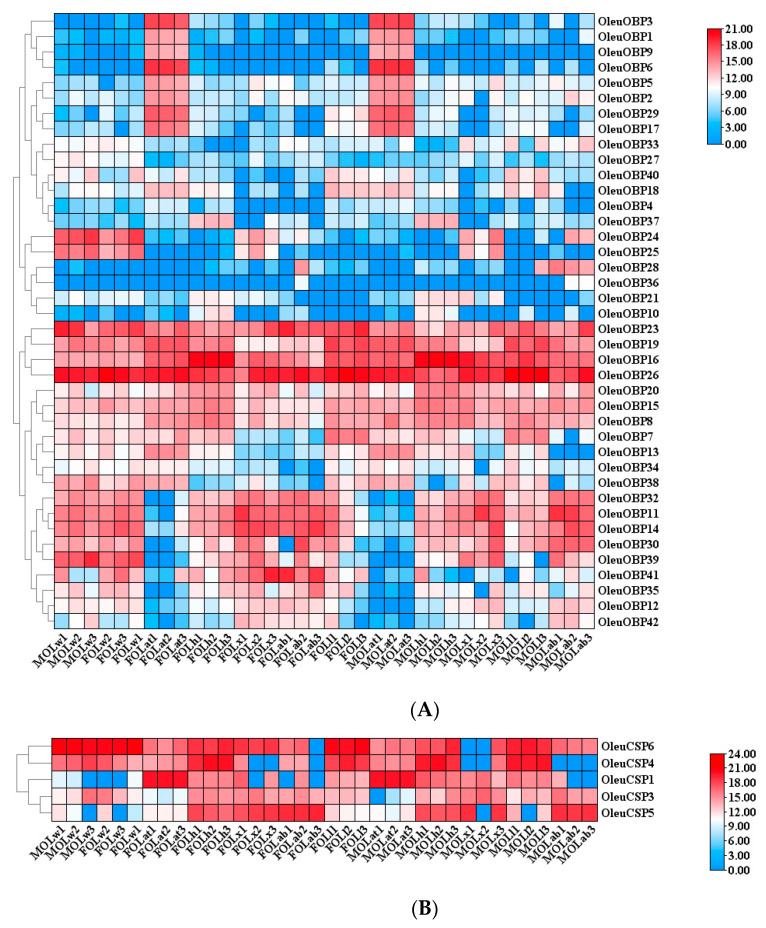
Expression profiles of chemosensory-related genes in *O. leucomelaena*. (**A**): OleuOBPs; (**B**): OleuCSPs; (**C**): OleuORs. OL: *O. leucomelaena*. FOLw: female wing; MOLw: male wing; FOLat: female antenna; MOLat: male antenna; FOLh: female head; MOLh: male head; FOLx: female thorax; MOLx: male thorax; FOLl: female leg; MOLl: male leg; FOLab: female abdomen; MOLab: male abdomen.

**Figure 3 ijms-25-09436-f003:**
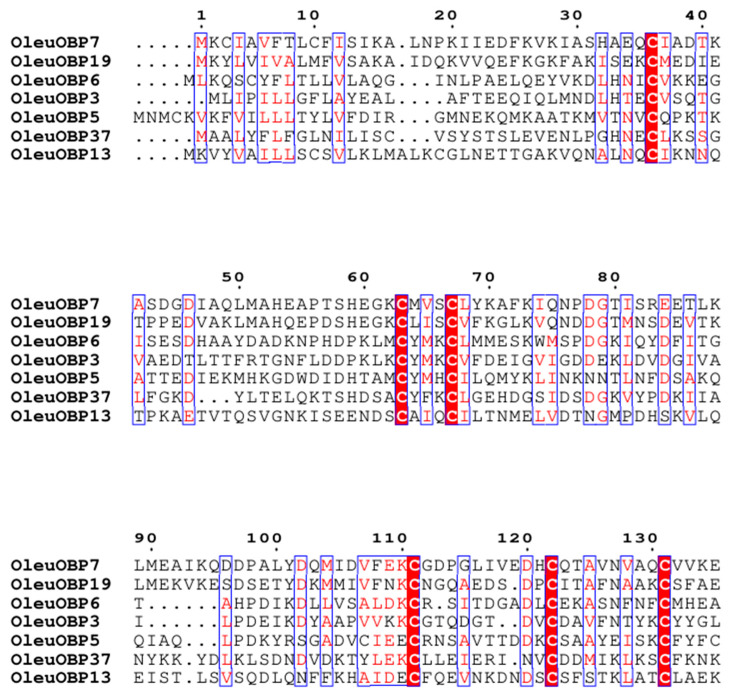
Alignment of the identified classic OleuOBPs. Conserved cysteine residues are selected and highlighted on a red background.

**Figure 4 ijms-25-09436-f004:**
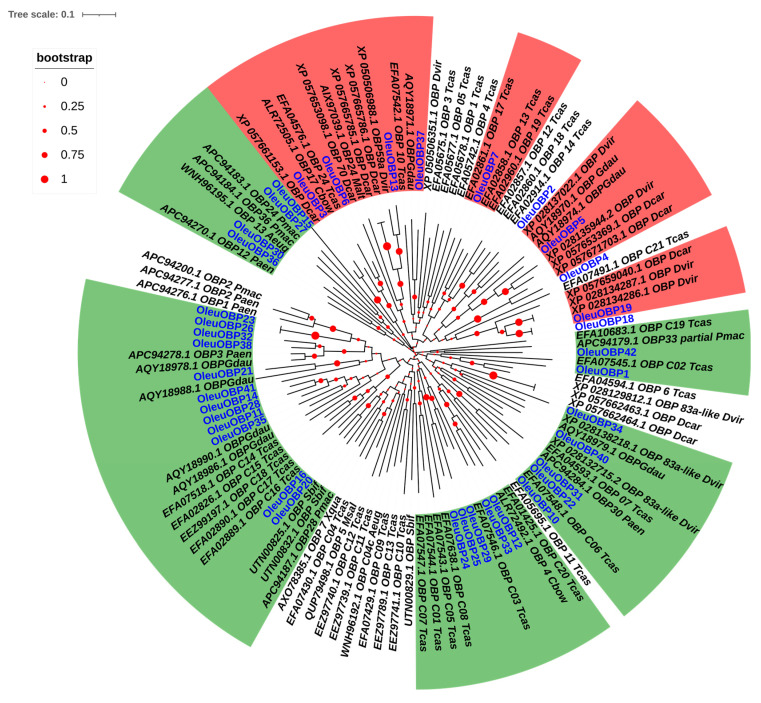
Neighbor-joining tree of candidate OBPs. Bootstrap values after 1000 replications. Oleu, *Oides leucomelaena*; Paen, *Pyrrhalta aenescens*; Pmac, *Pyrrhalta maculicollis*; Aeug, *Anthonomus eugenii*; Dcar, *Diorhabda carinulata*; Cbow, *Colaphellus bowringi*; Tcas, *Tribolium castaneum*; Malt, *Monochamus alternatus*; Dvir, *Diabrotica virgifera virgifera*; Gdau, *Galeruca daurica*; Pmac, *Pyrrhalta maculicollis*; Sbif, *Semanotus bifasciatus*; Msal, *Monochamus saltuarius*; Xqua, *Xylotrechus quadripes*. Red represents the classic OBP family and green represents the Minus-C OBP family.

**Figure 5 ijms-25-09436-f005:**
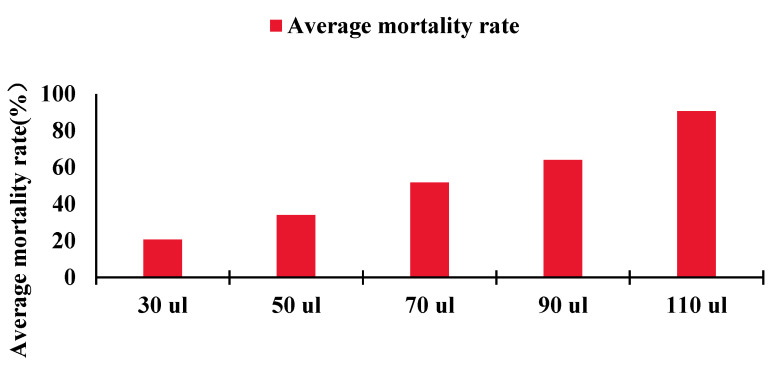
Concentration screening test results. X-axis indicates the dose of added compounds and Y-axis indicates the average mortality rate of the three groups.

**Figure 6 ijms-25-09436-f006:**
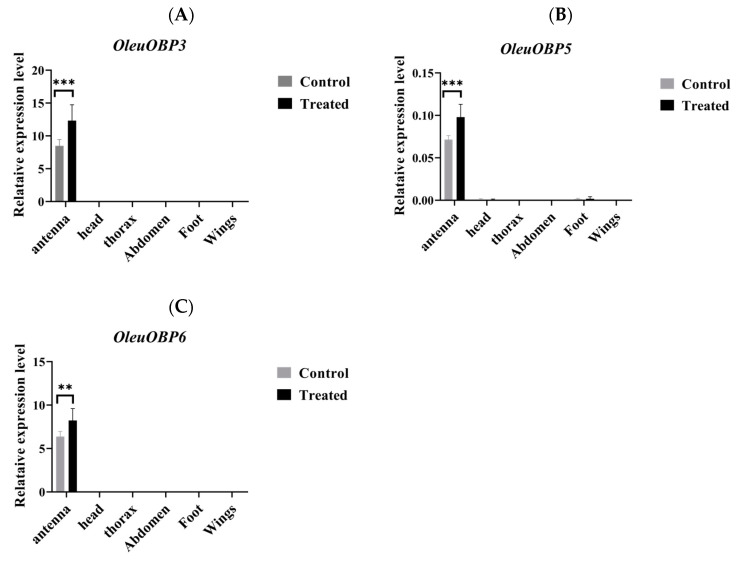
Analysis of OleuOBP gene expression after compound treatment. The standard error is represented by the error bar, and the ** indicates a very significant difference, and *** indicates an extremely significant difference. (**A**): OleuOBP3; (**B**): OleuOBP5; (**C**): OleuOBP6.

**Figure 7 ijms-25-09436-f007:**
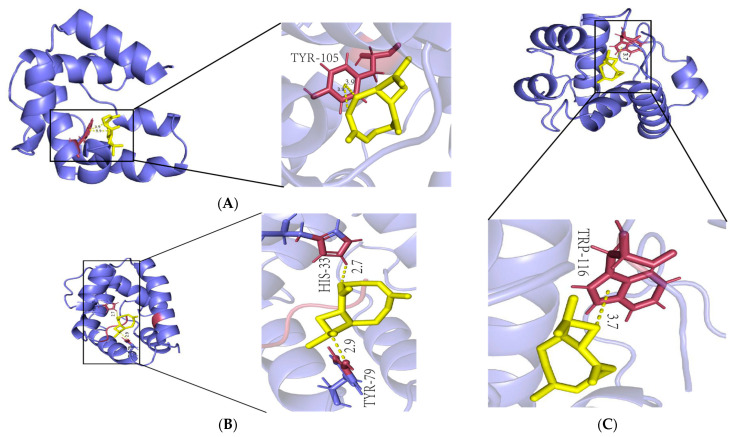
Docking pattern of protein and β-caryophyllene of OleuOBP. (**A**): OleuOBP3 and β-caryophyllene; (**B**): OleuOBP5 and β-caryophyllene; (**C**): OleuOBP6 and β-caryophyllene.

**Table 1 ijms-25-09436-t001:** Docking binding energy of OleuOBP to chemical compound molecules.

Protein	Ligand	Binding Energy (kJ/mol)
OleuOBP3	(Z)-anethole	−17.40544
OleuOBP5	(Z)-anethole	−20.69296
OleuOBP6	(Z)-anethole	−21.10336
OleuOBP3	β-caryophyllene	−24.12168
OleuOBP5	β-caryophyllene	−21.03352
OleuOBP6	β-caryophyllene	−31.7984
OleuOBP3	3-carene	−22.5936
OleuOBP5	3-carene	−17.9912
OleuOBP6	3-carene	−25.5224
OleuOBP3	4-allylanisole	−23.012
OleuOBP5	4-allylanisole	−16.736
OleuOBP6	4-allylanisole	−16.736

## Data Availability

All read data are available in the NCBI BioProject database under the project ID PRJNA1123008.

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
