# Peer review of "Molecular Characterization of Odorant-Binding Protein Genes Associated with Host-Seeking Behavior in Oides leucomelaena"

_ijms, 2024, doi:10.3390/ijms25179436_

Round 1

Reviewer 1 Report

Comments and Suggestions for Authors

This article focuses on the identification of chemosensory related protein genes in the chrysomelid pest Oides luecomelaena Weise that may play a role in the location of their economically important host Star Anise (Illicium verum). Identification of genes that are involved in host plant location will not only increase our fundamental understanding of the ecology of this pest but can additionally serve as the foundation for potential pest management strategies. The authors in this article identified 77 chemosensory related protein genes in this study, of which three were of particular interest. Odorant binding proteins (OBP) OweiOBP3, OweiOBP5, and OweiOBP6 were found to be antennal specific, suggesting their role in host plant location. Additionally, these OBPs also exhibited high binding energy to β-caryophyllene, a volatile component of the host plant blend, in a molecular docking experiment, further suggesting the role of these OBPs in host plant location. The authors contribute the first to our understanding of how O. luecomelaena can locate their host, of which current literature is scarce in describing this species. Additionally, this study provides a strong foundation for future functional studies to gain a better and more comprehensive understanding of the chemosensory system of O. luecomelaena.

1. Consider refraining from the title of the study by deleting the “functional”. This article provides evidence that suggests that these OBPs potentially play a role in host location in O. luecomelaena through the molecular docking analysis demonstrating binding affinity towards β-caryophyllene (major volatile constituent of the star anise volatile blend) and the fumigation toxicity bioassays showing an induction of OweiOBP3, 5, and 6 after treatment. However, no actual functional studies was performed to assess host seeking behavior and the potential relationship between the OBPs of interest and host plant volatiles.

2. Regarding homology modeling, the authors state that the determined three-dimensional structures of solved OBPs that had more than 30% homology were used for the molecular docking analysis, which will lead to the generation of a good model. Established literature (Venthur 2014 & Schwede et al. 2007) corroborates that this will lead to a reliable model to perform molecular docking analysis. However, Ginalski 2006 states that homology models with over 50% homology will lead to an even more reliable and precise model. I did not see the authors state the templates used for their molecular docking analysis, nor the homology scores of these templates to the models generated. I would suggest the addition of the protein data bank (PDB) four-digit codes of the templates used for the analysis, as well as the reporting of the homology of the templates to OweiOBPs 3,5, and 6.

3. I recommend that the Introduction and Discussion sections include citations to more recent review papers that cover the functions of odorant-binding proteins (OBPs) in host plant adaptation, insecticide resistance, and olfaction. For instance, citing Pelosi et al. (2018, Biological Reviews) and Abendroth et al. (2023, Frontiers in Insect Science) would provide a more comprehensive background and demonstrate the relevance of your work in the context of recent studies.

More details could be added to increase the clarity in several sections of the manuscript, as well as some of the figures. Specific examples can be found below, but a thorough check should be performed.

Specific comments:

1. Please enhance the resolution of Figures, specifically Figures 1, 2, 6, 7. These figures appear blurry and very hard to see.

2. Figure 5 is a bit confusing to understand without the help of reading additional text in the methods and/or results section. I would suggest increasing the clarity of the figure through the addition of the concentration next to the numbers in the figure (30, 50, 70, 90, 110) as well as an x-axis title describing the average mortality rates of each replication for each concentration. The Y-axis could be changed to the “average mortality rates”, as they are the averaged mortality of the beetles in one replication. Please rewrite the figure legend accordingly.

3. Page 8, Section 4.1: age of O. luecomelaena were not described. I understand that these beetles were collected from the field, but would it be possible for the authors to describe the time post collection of these beetles for the fumigation toxicity bioassays, if this is known?

4. Page 9, Section 4.4: How was the mortality of O. luecomelaena identified? Was there any mortality associated with the n-hexane control? It might be worth adding that no mortality was caused by the control, if this was found. Was there any difference in the mortality of male VS female O. luecomelaena?

Comments on the Quality of English Language

The manuscript should be proofread for small errors and typos

Reviewer 1 Report

  1. Consider refraining from the title of the study by deleting the “functional”. This article provides evidence that suggests that these OBPs potentially play a role in host location in O. luecomelaena through the molecular docking analysis demonstrating binding affinity towards β-caryophyllene (major volatile constituent of the star anise volatile blend) and the fumigation toxicity bioassays showing an induction of OweiOBP3, 5, and 6 after treatment. However, no actual functional studies was performed to assess host seeking behavior and the potential relationship between the OBPs of interest and host plant volatiles.

Response:We have revised the title of the article based on the reviewers' comments. “Molecular and Functional Characterization of Odorant-Binding Protein Genes associated with host-seeking behavior in the Oides leucomelaena Weise”

  1. Regarding homology modeling, the authors state that the determined three-dimensional structures of solved OBPs that had more than 30% homology were used for the molecular docking analysis, which will lead to the generation of a good model. Established literature (Venthur 2014 & Schwede et al. 2007) corroborates that this will lead to a reliable model to perform molecular docking analysis. However, Ginalski 2006 states that homology models with over 50% homology will lead to an even more reliable and precise model. I did not see the authors state the templates used for their molecular docking analysis, nor the homology scores of these templates to the models generated. I would suggest the addition of the protein data bank (PDB) four-digit codes of the templates used for the analysis, as well as the reporting of the homology of the templates to OweiOBPs 3,5, and 6.

Response: In our study, OBP5 and OBP6 showed more than 70% homology with the selected protein models, and OBP3 showed 31.58% homology. Corresponding information and model sources have been added in the Supplementary file.

  1. I recommend that the Introduction and Discussion sections include citations to more recent review papers that cover the functions of odorant-binding proteins (OBPs) in host plant adaptation, insecticide resistance, and olfaction. For instance, citing Pelosi et al. (2018, Biological Reviews) and Abendroth et al. (2023, Frontiers in Insect Science) would provide a more comprehensive background and demonstrate the relevance of your work in the context of recent studies.

Response: In the revised version, we have selected reviewer-recommended journals for presentation and discussion citations.

More details could be added to increase the clarity in several sections of the manuscript, as well as some of the figures. Specific examples can be found below, but a thorough check should be performed.

Specific comments:

  1. Please enhance the resolution of Figures, specifically Figures 1, 2, 6, 7. These figures appear blurry and very hard to see.

Response: The graphics and corresponding fonts in the article have been changed accordingly.

  1. Figure 5 is a bit confusing to understand without the help of reading additional text in the methods and/or results section. I would suggest increasing the clarity of the figure through the addition of the concentration next to the numbers in the figure (30, 50, 70, 90, 110) as well as an x-axis title describing the average mortality rates of each replication for each concentration. The Y-axis could be changed to the “average mortality rates”, as they are the averaged mortality of the beetles in one replication. Please rewrite the figure legend accordingly.

Response: Based on the reviewer's comments, we modified Figure 5 by changing the Y-axis to the average mortality rate, and on the X-axis we chose to add specific units, and in conducting the experiments, we clarified that different treatment gradients were carried out by adjusting the amount of additions, and that the other environmental conditions were kept the same in order to ensure the scientific validity of the study.

  1. Page 8, Section 4.1: age of O. luecomelaena were not described. I understand that these beetles were collected from the field, but would it be possible for the authors to describe the time post collection of these beetles for the fumigation toxicity bioassays, if this is known?

Response: Insects used for fumigation experiments were collected from the collection in April 2024, adult white snakeheads were selected, collected and centrally reared in the laboratory for three days before being subjected to fumigation experiments, which lasted for one hour of treatment

  1. Page 9, Section 4.4: How was the mortality of O. luecomelaena identified? Was there any mortality associated with the n-hexane control? It might be worth adding that no mortality was caused by the control, if this was found. Was there any difference in the mortality of male VS female O. luecomelaena?

Response: O. leucomelaena was removed for observation and examined under an in vitro microscope with a clean brush and forceps; no response to touch was considered dead, and then the white snakeheads were observed for survival and mortality was calculated. Prior to conducting the experiment, we performed a preexperiment with hexane treatment, which by itself resulted in low mortality and no male or female differences. Instead, we chose hexane to dilute the mixtures in our experiments with lower additions.

Reviewer 2 Report

Comments and Suggestions for Authors

This article is devoted to the study of the genes of proteins that determine the behavior of beetles in response to olfactory stimuli. This is a very interesting and fundamental study. The experiment is well designed. The results of the article supplement current knowledge about the perception of odors by insects. After eliminating technical deficiencies, this article can be recommended for publication.

  1. Line 4: I recommend deleting the word "Weise" and after the Latin name of the species. In brackets, the Latin name of the order and family should be added. This will increase the number of citations to this article from the Scopus and Web of Science databases.
  2. Lines 13 and 37: after the Latin name of the species, in brackets, the Latin name of the order and family should be added). There is no need to duplicate this text "(O. leucomelaena Weise)". After the word "Weise" on this line, the year of the species description in accordance with the International Code of Zoological Nomenclature should be indicated.
  3. Starting from line 43 and further throughout the article, the word "Weise" should be deleted. 4. Lines 113, 139, 163, 182: readers will not see anything in these figures. The figure needs to be resized so that the size of all the letters and numbers in the figure is approximately equal to the size of the letters and numbers in the text of the article.
  4. Lines 120, 125, 128, 161 and many others: any first mention of any animal in this article, in accordance with the International Code of Zoological Nomenclature, must be mentioned by the author's last name and year. It is also advisable to add the name of the order and family after this in brackets, separated by commas.
  5. Line 179: the figure is not formatted according to the rules, using a combination of a table and a histogram. This is unacceptable in professional articles in prestigious scientific journals. Format this data as a table. It is necessary to apply a correct method of multiple comparison of samples in this table, for example, the Tukey test.
  6. Line 182: Columns for insect antennae should be placed in one coordinate system, and columns for other parts of the beetle's body should be placed in a different coordinate system. The height of the columns should be increased so that readers can see the standard error on all columns, as well as the reliability of the differences.
  7. Line 279: Material and methods should be placed before the Results section. This will affect the numbering of references.
  8. In the list of references, authors randomly write words with a capital or lowercase letter. This is unacceptable. In the titles of articles, only proper names are capitalized. In the titles of journals, all words are capitalized, except for articles, prepositions, and conjunctions. DOI indexes should be added everywhere, where they exist.

  1. Line 4: I recommend deleting the word "Weise" and after the Latin name of the species. In brackets, the Latin name of the order and family should be added. This will increase the number of citations to this article from the Scopus and Web of Science databases.

Response: The word “Weise” has been deleted from the text and the Latin names of the orders and families have been added.

  1. Lines 13 and 37: after the Latin name of the species, in brackets, the Latin name of the order and family should be added). There is no need to duplicate this text "(O. leucomelaena Weise)". After the word "Weise" on this line, the year of the species description in accordance with the International Code of Zoological Nomenclature should be indicated.

Response: Changes have been made accordingly.

  1. Starting from line 43 and further throughout the article, the word "Weise" should be deleted.

Response: Changes have been made accordingly.

  1. Lines 113, 139, 163, 182: readers will not see anything in these figures. The figure needs to be resized so that the size of all the letters and numbers in the figure is approximately equal to the size of the letters and numbers in the text of the article.

Response: Changed the article accordingly to address the issue of images in the article.

  1. Lines 120, 125, 128, 161 and many others: any first mention of any animal in this article, in accordance with the International Code of Zoological Nomenclature, must be mentioned by the author's last name and year. It is also advisable to add the name of the order and family after this in brackets, separated by commas.

Response: Corresponding authors and years have been added to the names of the insects in the study.

  1. Line 179: the figure is not formatted according to the rules, using a combination of a table and a histogram. This is unacceptable in professional articles in prestigious scientific journals. Format this data as a table. It is necessary to apply a correct method of multiple comparison of samples in this table, for example, the Tukey test.

Response: The graphs have been modified accordingly and the X and Y coordinates of the images have been changed to refer to the format of the published article.

  1. Line 182: Columns for insect antennae should be placed in one coordinate system, and columns for other parts of the beetle's body should be placed in a different coordinate system. The height of the columns should be increased so that readers can see the standard error on all columns, as well as the reliability of the differences.

Response: In response to this question, which was discussed at length, we identified selected OBPs as tentacle-specific OBPs that have very low or no expression in other tissues and for which qPCR experiments were performed when low gene expression was detected in other tissues. In addition, graphical combinations were chosen to highlight their tentacle specificity.

  1. Line 279: Material and methods should be placed before the Results section. This will affect the numbering of references.

Response: When writing according to the journal format, the materials and methods are organized at the end of the article.

  1. In the list of references, authors randomly write words with a capital or lowercase letter. This is unacceptable. In the titles of articles, only proper names are capitalized. In the titles of journals, all words are capitalized, except for articles, prepositions, and conjunctions. DOI indexes should be added everywhere, where they exist.

Response: The reference format has been completely revised.